

# Dynamics of the snow grain size in a windy coastal area of Antarctica from continuous in-situ spectral albedo measurements

Sara Arioli[1], Ghislain Picard[1], Laurent Arnaud[1], and Vincent Favier[1]

[1]Université Grenoble Alpes, CNRS, IRD, Grenoble INP, IGE, 38000 Grenoble, France

**Correspondence:** Sara Arioli (sara.arioli@univ-grenoble-alpes.fr)

**Abstract.** The grain size of the superficial snow layer is a key determinant of the surface albedo in Antarctica. Its evolution is the result of multiple interacting processes, such as dry and wet metamorphism, melt, snow drift and precipitation. Among them, snow drift has the least known and least predictable impact. The goal of this study is to relate the variations of surface snow grain size to these processes in a windy location of the Antarctic coast. For this, we retrieved the daily grain size from 5

year-long in-situ observations of the spectral albedo recorded by a new multi-band albedometer, unique in terms of autonomy and described here for the first time. An uncertainty assessment and a comparison with satellite-retrieved grain size were carried out to verify the reliability of the instrument and an RMSE up to 0.16 mm on the observed grain size was found. By relating these in-situ measurements to timeseries of snow drift, surface temperature, snow surface height and snowfall, we established that the evolution of the grain size in the presence of snow drift is complex and follows two possible pathways: 1) A decrease in

the grain size (about half of our measurements) resulting from the deposition of small grains advected by the wind. Surprisingly, this decreases is often (2/3 of the cases) associated with a decrease of the surface height, i.e. a net erosion over the drift episode, 2) an increase of the grain size (the other half) either due to the removal of the surface layer, or metamorphism. However, we note that this increase is often limited with respect to the increase predicted by a theoretical metamorphism model, suggesting that a concomitant deposition of small grains is likely. At last, we found that wind also completely impedes the deposition of

snowfall during half of the observed precipitation events. When this happens, the grain size evolves as if precipitation was not occurring. As a result of all these processes, we conclude that the grain size in a windy area remains more stable than it would be in the absence of snow drift, hence limiting the variations of the albedo and of the radiative energy budget.

## 1   Introduction

Snow covered areas have a cooling effect on the climate both at local and global scale because of their high albedo (Zhang

et al., 2022). In Antarctica, as the sun is above the horizon for most of the day during summer, the surface receives substantial amounts of shortwave radiation (Van Den Broeke et al., 2004; Bai et al., 2022). Thus, even small fluctuations in the surface albedo cause important variations in the local surface energy budget. In areas of pristine snow, which is often the case in Antarctica (Warren et al., 2006), the main determinant of the snow albedo is the size of the snow grains that constitute the superficial layer ($\approx$1 cm thick). It is a highly dynamical and complex variable, as its evolution is the result of many competing

processes and of the recent history of the snowpack.



Firstly, metamorphism transforms the snow grains over time, driven by the snow temperature, temperature gradient and water liquid content. It generally increases the size of the snow grains following two possible pathways depending on whether the snow is dry or wet. Dry metamorphism occurs for temperatures below 0°C and induces a moderate increase of the grain size whose rate increases with temperature and temperature gradient (Colbeck, 1982). In contrast, wet metamorphism leads

to a fast growth of the snow grains that greatly depends on the liquid water content (Marsh, 1987; Brun et al., 1992). Further coarsening occurs when melting snow freezes during the night, forming "melt-freeze crusts" (Colbeck, 1973). In any case, the snow grains' growth induces an albedo decrease and an enhancement of the sunlight absorption. Metamorphism, whether dry or wet, is always active and the induced grain size increase is irreversible.

Snowfall, on the other hand, is an intermittent process that replaces the superficial snow layer by depositing fresh snow.

The grain size of the deposited snow depends on the atmospheric conditions during precipitation and is usually small (Domine et al., 2007; Walden et al., 2003) compared to the evolved snow on the ground. Thus, such deposition events over coarse-grained snow effectively increase the snow surface albedo. For instance, Picard et al. (2012) observed an albedo increase of 0.03 throughout the summer season compared to the average for years with higher summer precipitation at Dome C, on the Antarctic Plateau. The whitening effect of snowfall on the surface also inhibits surface melt in areas with frequent precipitation

with respect to areas that are subject to long dry periods (Jakobs et al., 2021).

The transport of snow operated by the wind, commonly referred to as snow drift, is another intermittent process that plays a more complex and less predictable role. Generally, wind lifts snow grains when its speed exceeds a threshold that depends on the cohesion of the surface snow (Kind, 1986). This process is more effective after snowfall, when the cohesion of the superficial layer is lowest, and weaker on dense, old surfaces such as melt-freeze crusts (Pomeroy and Gray, 1990; Lenaerts

and Van den Broeke, 2012). Once advected from the surface, the snow grains either creep, fall following a ballistic trajectory – saltation – or are suspended and transported by the wind over long distances – turbulent diffusion – (Déry and Yau, 1999; Barral et al., 2014). During their transport, the snow grains are fractured by the impact with the surface or other drifting grains and shrink, as they partially or totally sublimate in the unsaturated air (Amory and Kittel, 2019). In this context, snow drift modifies the properties of the snow surface in two possible ways that lead to opposite effects on the surface albedo. On the one

hand, the erosion generally uncovers old snow layer and may result in an increase of the grain size at the surface, which lowers the albedo and may enhance surface melt (Bintanja, 1999; Lenaerts et al., 2016). On the other hand, the deposition of drifting snow grains, whose size is reduced by fragmentation and sublimation, has a similar effect to snowfall (Domine et al., 2009).

All these processes interplay according to their frequency and magnitude, driving the evolution of the snow grain size, and therefore of the surface albedo, at different time scales. As a result, the effect of these interactions between metamorphism and

these meteorological phenomena on the resulting surface grain size is complex and location specific. Yet, the impact of snow drift on the optical properties of the snow surface has been scarcely investigated. Few studies addressed the extreme case of blue ice areas (Winther et al., 2001; Lenaerts et al., 2016) but, to our knowledge, no previous study focused on the impact of snow drift on the snow grain size over perennial snow covers, left alone its interaction with dry metamorphism, surface melt and snowfall.



The goal of this study is to relate the variations of the grain size deduced from the observed surface albedo to the processes of snowfall, melt and drift in a windy location of the Antarctic coast. More specifically, we aim at identifying the prevailing processes occurring (1) in the presence of snow drift, (2) in the presence of snowfall, (3) in the presence of surface melt.

To achieve this goal, we exploit multi-year timeseries of the snow grain size obtained from a new, automated albedometer with multiple spectral bands that was installed at two locations in Adélie Land, East Antarctica. This instrument is described
here in detail for the first time. The study area, the instrumentation available and the processing of raw albedo measurements are described in Section 2. The results of the processing, the retrieved time series of the grain size, and the comparison with local meteorological parameters are detailed in Section 3. The discussion and conclusion are in Section 4 and 5 respectively.

## 2   Materials and methods

### 2.1   Study area and meteorological measurements

The study area is located in the coastal area of Adélie Land in East Antarctica, on the ice sheet, in front of the Dumont d'Urville station located on Petrel island. Two specific locations are considered (Figure 1). The point D5 (66.70°S, 139.88°E) is located 1.5 km from the shoreline at 170 m a.s.l., and the point D17 (66.72°S, 139.72°E) at 6 km inland and 415 m a.s.l.. Air temperature in D5 frequently exceeds 0°C during daytime in summer, while winter minima rarely reach less than -30°C. In D17, summer maxima are slightly below 0°C and winter minima around -30°C. The area is characterized by strong episodic
katabatic winds blowing from the Antarctic Plateau to the coast (Wendler et al., 1997) that affect the two locations. These winds are responsible of snow erosion and deposition, carving sastrugi at the surface. Blowing snow events are common, as reported by expeditioners (Amory, 2020). D17 is equipped with an Automated Weather Station (AWS) that includes sensors for temperature, wind speed and relative humidity at 2 m height, a 4-flux radiometer (Kipp & Zonen CNR4 sensor) for the measurement of the incoming and outgoing shortwave and longwave radiation, two ultrasonic distance sensors (Campbell
Scientific SR50) for the measurement of the snow surface height, and a FlowCapt[TM] instrument (Chritin et al., 1999) for the estimation of the snow drift volume.

### 2.2   The Multiband instrument

In addition to common meteorological equipment, two home-made spectro-albedometers called Multiband were deployed in January 2017 at D5 and D17 to monitor the snow spectral albedo (Figure 2).
The Multiband instrument is a multi-spectral band optical radiometer operating in the visible, near and short wave infrared domains. It runs autonomously in the harsh conditions of Antarctic environment thanks to the combination of a solar panel and a battery, with interventions needed just to recover measurements, once a year. It measures the incoming and reflected radiation in order to provide the spectral albedo of the snow surface. The structure of the Multiband instrument is depicted in Figure 3. Incident and reflected solar radiation are collected by two home-made cosine-shaped light collectors located on the
up and down-facing sides of the instrument's head following Picard et al. (2016). Two 6 m long optical fibers (800 μm core





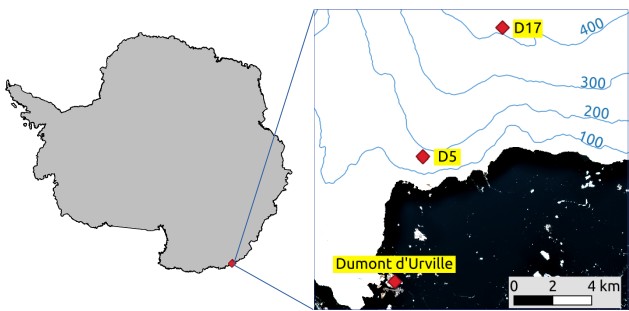

**Figure 1.** Location of the study area with the sites D5 and D17, in the vicinity of the French station of Dumont d'Urville in Adélie Land, East Antarctica.

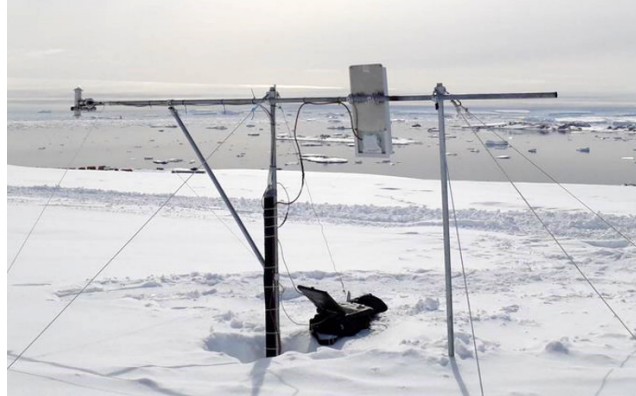

**Figure 2.** Picture of the Multiband albedometer installed in D5, Adélie Land. The measurement head is on the left.

diameter, QP800-VIS NIR, Ocean Optics) transmit the radiation inside the instrument, to a rotating filter wheel containing 12 optical filters at user-selected wavelengths. The selection for this study was optimized for the albedo of snow and includes only 7 different wavelengths. They are 500, 600, 800, 925, 1050, 1300 and 1550 nm (TECHSPEC® Hard Coated OD 4.0 25 nm Bandpass Filters, Edmund Optics). During a measurement cycle, the wheel rotates by successive steps and the radiance

is measured for both the incident and reflected light channels by two Dual Band Si/InGaAs photodiodes (DSD2, Thorlabs). The measurements of the radiance are simultaneous for the two channels, but the photodiodes are located in front of different slots of the filter wheel, that means that the order of measurement with respect to the wavelength is shifted between the two channels. For this reason, the remaining slots of the filter wheel were filled with duplicates of the filters at 500, 800, 1050 and 1300 nm in order to make some of the measurements for the same wavelength simultaneous for the two channels. Table

1 shows the resulting list of wavelengths of the selected filters and the order of measurement during the rotation of the filter wheel. The measurements chosen for the albedo retrieval are shaded in gray. The rotation cycle of the filter wheel lasts nearly 30 s. A high-gain, electronic board converts and amplifies the signal from the two dual photodiodes. Then, a datalogger registers





the measurements (CR1000, Campbell Scientific). Since the incident and reflected measurements for 3 wavelengths are not simultaneous, we record in addition the changes of the incoming radiation during the 30 s of the measurement cycle. To this

end, another upward-facing optical fiber located near the albedometer head transmits the environmental incoming radiation to a Si photodiode (FDS100, Thorlabs) without any spectral filter. This photodiode acquires one wideband measurement (350-1100 nm) for each step of the wheel, that is converted, amplified and recorded as the narrow-band measurements. At last, once per cycle, the dark current is measured for each dual photodiode using an obstructed window in one of the filter wheel slots. After suited artifact corrections detailed in Section 2.3, the spectral albedo is computed as the ratio of the reflected and incident

irradiance spectra. The albedometer head was installed at the edge of a North-South oriented metal gantry (fig. 2) in order to minimize gantry components or shades in its field of view. The head height at the moment of installation was of 1.6 m above the surface, which decreased on average by 18 cm per year. After installation, the head was leveled thoroughly. The active sensors, batteries and acquisition electronics are buried in the snow to grant the temperature stability of the electronics. The Multiband in D5 and D17 have been running during Antarctic summer from January to March 2017 and then uninterruptedly

from December 2019 until today. Interventions were carried out once a year to download observations and to check the leveling of the albedometer head. The most recent one occurred in December 2021.

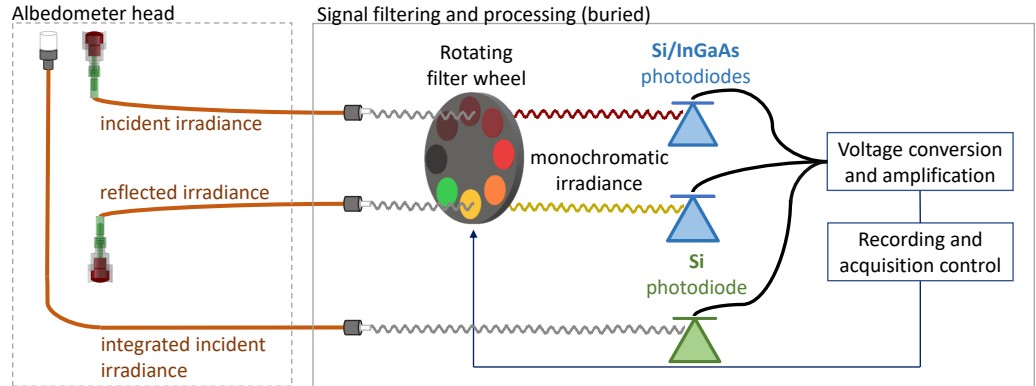

**Figure 3.** Scheme of the Multiband albedometer displaying the albedometer head (dashed frame) connected through optical fibers (in orange) to the buried filtering and processing electronic systems (solid frame).

## 2.3 From raw irradiance measurements to calibrated albedo

A Multiband measurement cycle consists of 7 raw incident and reflected irradiance spectral measurements, 4 dark values, one for each band of the two dual band photodiodes, as well as 12 wideband irradiances. Raw albedo measurements are subject

to multiple artifacts and errors that need to be corrected. We developed a series of processing steps to derive accurate albedo spectra as shown in Figure 4. These steps are described here.

    **Raw measurement selection**. Measurements acquired under unstable or low-sun illumination conditions are first discarded. These conditions are detected when the standard deviation of the 12 wideband measurements over the 30 s cycle exceeds 1%



| Order | Incident | | | Reflected | | |
|---|---|---|---|---|---|---|
| | $\lambda$ (nm) | Si | InGaAs | $\lambda$ (nm) | Si | InGaAs |
| 01 | Dark | ✓ | ✓ | Dark | ✓ | ✓ |
| 02 | 500 | ✓ | | 500 | ✓ | |
| 03 | 800 | ✓ | | 800 | ✓ | |
| 04 | 1050 | ✓ | | 1050 | ✓ | |
| 05 | 1300 | ✓ | | 1300 | | ✓ |
| 06 | 500 | ✓ | | 600 | ✓ | |
| 07 | 800 | ✓ | | 925 | ✓ | |
| 08 | 1050 | ✓ | | 1550 | | ✓ |
| 09 | 1300 | | ✓ | 500 | ✓ | |
| 10 | 600 | ✓ | | 800 | ✓ | |
| 11 | 925 | ✓ | | 1050 | ✓ | |
| 12 | 1550 | | ✓ | 1300 | | ✓ |

**Table 1.** List and order of the radiance measurements performed by the two Dual Band Si/InGaAs photodiodes during the rotation of the filter wheel for the incident and reflected channels. The check mark identifies the band of the photodiode used for each wavelength $\lambda$. Measurements in a same row are simultaneous, and cycling over the 12 rows takes about 30 s. The chosen ones for the albedo retrieval are shaded in gray.

of the mean wideband radiance. Measurements collected with Solar Zenith Angle (SZA) larger than 70° are also discarded
because of the reduced performance of the collectors at grazing angles (Picard et al., 2016), and the lower interest for the surface energy budget. Finally, we had to exclude the irradiance measurements at 1550 nm because of their poor signal to noise ratio.

**Dark current**. The raw radiance measurements are corrected to compensate for the dark current of the photodiodes. The dark current measurements recorded once every measurement cycle are therefore subtracted from the spectral irradiance values
according to the corresponding channel and photodiode band.

**Cross calibration**. The difference of transmittivity through the cosine collectors and the fibers, as well as the difference of sensitivity of the electronics between the incident and reflected channels, are compensated. For this, a specific experiment, called cross-calibration (CC), was carried out during the initial deployment of the instrument. During this experiment, the light collectors of the two channels (incident and reflected) were positioned side by side and leveled looking upward so that
they both receive exactly the same irradiance. Spectral measurements were acquired during two days in this configuration, for both channels, every hour during daytime and every 10 minutes in the hour following solar noon. These CC acquisitions were then processed as follows: the acquisitions during unstable illumination conditions were first discarded and the remaining





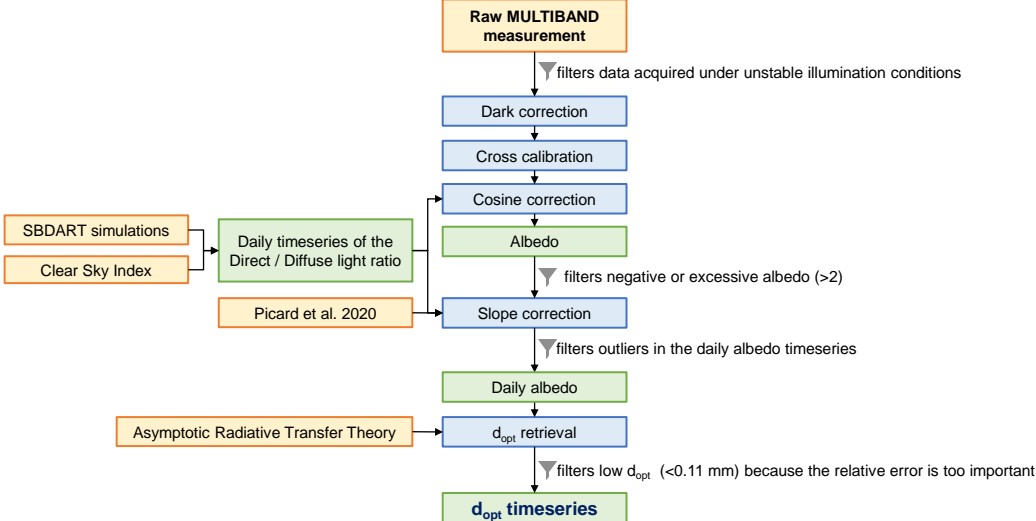

**Figure 4.** Workflow of the processing showing input data (orange), processing steps (blue) and intermediate and final results (green). The funnel sign indicates filtering of data between two steps.

were corrected for dark and averaged for each wavelength and channel. The ratio, calculated for each wavelength, between the averaged CC reflected irradiance and the averaged CC incident irradiance gives the relative difference of sensitivity between
the two channels at this specific wavelength. Then, to compensate for the small differences between the two channels during normal operation, this ratio is multiplied to the incident channel spectra throughout the timeseries.

**Cosine response correction**. For an accurate measurement of the irradiance, the ideal light collector has a perfect cosine response. Our home-made collectors are identical to those designed by Picard et al. (2016) and their response is typically better than 6% in the incidence angle range 0–70°, while it degrades for more grazing angles. Despite these excellent optical
performances and in order to improve the albedo accuracy, the residual error is corrected by an ad hoc processing step following the procedure described in Picard et al. (2016). Shortly, the collector response was first measured in the lab by illuminating the collector with a collimated beam at different incident angles and wavelengths. This response is used to normalize the actual measurements during normal operations in the field. However, this normalization must be applied to the direct component of the incident radiation only, neither the diffuse incident component nor the reflected component (which is fully diffuse) are
subject to this artifact. This implies to be able to partition the direct and diffuse incident irradiance for each measurement at every wavelength. For this, we rely on numerical simulations instead of on additional measurements as described in the next point.

**Direct to diffuse irradiance ratio**. The direct / diffuse partition in the incident irradiance was computed from simulations with the Santa Barbara DISORT Atmospheric Radiative Transfer (SBDART) model (Ricchiazzi et al. (1998)). The simulations
considered a standard sub-arctic winter atmosphere – the closest to summer Antarctic coastal conditions – pristine atmosphere and no clouds. The coordinates and elevation of the simulated fluxes are (66.69°S, 139.90°E) and 90 m a.s.l..





**Clear Sky Index**. The cloud cover is a major determinant of the direct/diffuse partition. For the sake of simplicity, we classify each Multiband measurement as realized under either clear or overcast sky conditions. For this purpose, the Clear Sky Index (CSI) proposed by Marty and Philipona (2000) was computed for all measurements, using the incident long-wave irradiance, 2-m air temperature and relative humidity from the AWS measurements located at D17. Briefly, Marty and Philipona (2000) consider as clear sky the condition CSI<1. However, we choose to classify as clear sky all measurements acquired with CSI≤1.25, as we consider a thin or partial cloud cover to be better represented by clear sky conditions than overcast. Finally, days are classified as clear sky when at least 75% of measurements acquired during that day are clear sky, overcast otherwise. Clear sky days are assigned the direct/diffuse partition modeled with SBDART, while overcast days are considered as full diffuse for each wavelength.

**Slope correction**. Sensor tilt and surface slope alter the measured albedo by increasing or decreasing the amount of radiation that hits the collectors as the solar azimuth angle varies throughout the day (Grenfell et al., 1994). These impact the measurement of albedo by increasing or decreasing the amount of the measured reflected radiation according to the slope orientation with respect to the sun's position. For instance, Picard et al. (2020) shows that an albedo error of 0.01 is detectable for slopes as small as 0.6° when the solar zenith angle is 45°. Moreover, the magnitude of the error increases with the solar zenith angle and with the slope magnitude. However, it decreases with the cloud cover until being negligible during overcast days (Weiser et al., 2016; Bogren et al., 2016). For the Multiband instruments installed at D5 and D17, the sensor tilt was found to be negligible, <0.5°, thanks to a thorough leveling of the albedometer head carried out during installation and monitored during the annual servicing of the instruments. We therefore exclude significant tilt. Besides, with an installation height of 1.6 m, the collected light is coming from a circular area of a few meters in diameter on the surface, so that local slopes and roughness are always present and significant at this scale. Furthermore these slopes are variable in time at D5 and D17 due to the presence of episodic sastrugi and other bedforms. To correct for the presence of slope, we apply one of the methods described in Picard et al. (2020), "the constrained correction of the diurnal cycle of albedo with unknown slope parameters". The method is qualified of "constrained" because it assumes that the corrected diffuse albedo has a fixed mean value between 500 nm and 600 nm (0.98 by default). Briefly, the correction is applied every day independently and assumes 1) that the true snow diffuse albedo is constant during the day (i.e. snow properties are not changing), 2) that the daily cycle of measured albedo is ascribable to the unknown surface slope artifacts and the known solar zenith angle variations only, and 3) that the slope is planar at the scale of the sensor footprint. The output for every day consists in a) a single spectrum of corrected diffuse albedo, b) the values of the slope inclination and aspect and c) the full timeseries of modeled albedo spectra for each measurement day.

## 2.4 SSA retrieval

The Specific Surface Area (SSA, $m^2kg^{-1}$) and the effective optical grain diameter ($d_{opt}$, mm) are retrieved from the diffuse albedo spectrum for each day. The first one is the surface area of the ice-air interface of snow per unit of mass of ice. The second is the diameter of a collection of spheres having the same SSA as the actual snow. These variables are interchangeable





and linked by the following relationship:

$$d_{opt} = \frac{6}{\rho_{ice} \cdot SSA} \tag{1}$$

where $\rho_{ice}$=917 $\mathrm{kg\,m^{-3}}$ is the ice density. Both variables are retrieved with the two-parameter model described in Picard et al. (2016) and derived from the analytical Asymptotic Radiative Transfer theory (ART; Kokhanovsky and Zege, 2004). Briefly, the direct and diffuse albedo are modeled as:

$$\alpha^{dir}(\lambda) = A \cdot exp\left(-\frac{12}{7}(1 + 2cos\theta)\sqrt{\frac{2B\gamma(\lambda)}{3\rho_{ice}SSA(1-g)}}\right) \tag{2}$$

$$\alpha^{diff}(\lambda) = A \cdot exp\left(-4\sqrt{\frac{2B\gamma(\lambda)}{3\rho_{ice}SSA(1-g)}}\right) \tag{3}$$

where $\lambda$ is the wavelength, $A$ is an unknown scaling factor that accounts for wavelength-independent artifacts (residual calibration error or illumination fluctuations) and $\theta$ is the solar zenith angle. Here we take the absorption enhancement parameter $B$=1.6 and the asymmetry factor $g$=0.85 as suggested by Libois et al. (2014). $\gamma(\lambda)$ is the ice absorption taken from Warren and Brandt (2008). This model is valid for albedo measured on a flat surface, vertically and horizontally homogeneous snowpack and clean snow, which is the case in our area (after the slope correction). The two unknown parameters, $A$ and SSA are computed by fitting the model to the observed diffuse albedo at 800, 925 and 1050 nm using a least-square minimization. The effective optical grain diameter is finally deduced from SSA using Equation 1.

## 2.5 Additional filtering

Over 10,000 measurement cycles were acquired at each location (every hour during daytime and every 10 minutes in the hour following local noon, 3:00 to 4:00 UTC), from September to March between January 2017 and December 2021. For D5 and D17 respectively, 9.6% and 20.1% of measurements cycles were discarded because of unstable illumination conditions during the cycle. An additional 0.6% and 0.5% of the measurements were discarded after dark subtraction, inter-calibration and cosine correction because the albedo computed before slope correction contained values below 0 or above 2 for at least one wavelength. A negative albedo value suggests an incorrect dark correction (signal smaller than the dark), while values above 2 are possibly due to large slopes, considered too difficult to correct (Picard et al., 2020), or to snowfall obturating the upward-looking collector. The slope correction was then applied to the remaining spectra, for each of the over 600 days at each station. To assess the quality of the fit, we computed the Root Mean Square Error (RMSE) between all the measured and modeled albedo spectra acquired during each day and filtered out the outlier acquisitions by using a z-score test with a $2\sigma$ threshold. The slope correction was then repeated without these outliers. After this step, if the RMSE>0.05, the output of the fit is definitively discarded (14.6% and 15.0% of the days at D5 and D17 respectively). This step helps identifying and removing the measurements that may have been acquired under different sky conditions than those assigned to the whole day.

Over 600 slope-corrected, daily spectral diffuse albedo were obtained for each station. The average RMSE is 0.021 and 0.020 for D5 and D17 correspondingly. To assess how such an error impacts the retrieved SSA and $d_{opt}$, we computed the theoretical





diffuse albedo of five SSA values between 10 and 60 using Eq. 3 (with $A$=1) and added a normally-distributed random noise
with a standard deviation of 0.021. SSA and $d_{opt}$ were then retrieved for each of 1,000 perturbed albedo spectra for each of
the five SSA values. The resulting distributions are shown in Figure 5a for the SSA and Figure 5b for the optical diameter. The

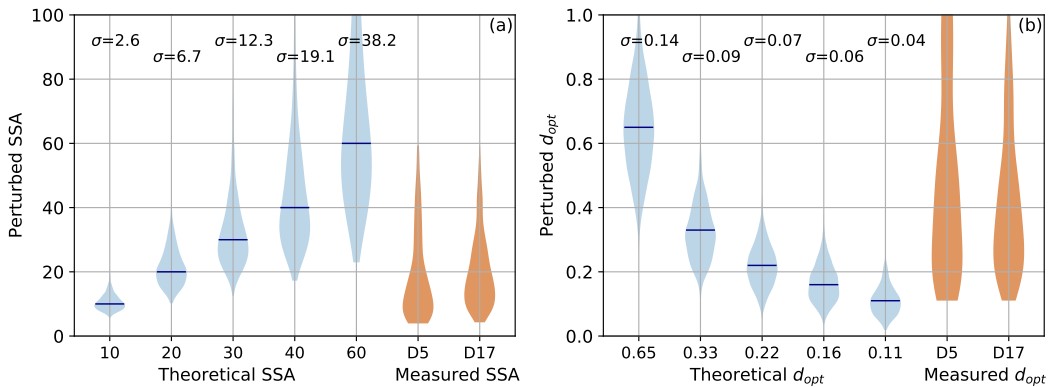

**Figure 5.** Error induced on the SSA (a) and on the $d_{opt}$ (b) from an error of 0.021 on the measured albedo (in blue) and distributions of the
observed values (in orange).

standard deviations of the perturbed distributions increases with the SSA, from 26% for a SSA of 10 $\mathrm{m^2kg^{-1}}$ to 64% for a SSA
of 60 $\mathrm{m^2kg^{-1}}$ with respect to the true value. The distributions of the perturbed $d_{opt}$ in Figure 5b show a decreasing absolute
standard deviation but an increasing relative standard deviation as $d_{opt}$ decreases. As a conclusion, we consider that the error
for measured SSA above 60 $\mathrm{m^2kg^{-1}}$ – and the corresponding $d_{opt}$ of less than 0.11 mm – are too large for our application,
and these measurements are discarded. The final results account for 541 and 601 days with valid SSA and $d_{opt}$ for D5 and D17
correspondingly. Their final distributions are shown in Figure 5a and 5b (in orange).

## 2.6 Evaluation against satellite and in-situ observations

As no reference in-situ measurements of grain size are available at our site, we first rely on satellite observations for an
independent assessment of the quality of our retrieval and, second, on broadband albedo measurements recorded at D17 using
a conventional shortwave radiometer (CNR4).

The measured $d_{opt}$ are compared to $d_{opt}$ retrieved from images from the OLCI instrument (Ocean and Land Colour In-
strument, on board of ESA's Copernicus Sentinel-3A/B, Nieke et al., 2012). The retrieval is done using the algorithm SICE,
the "Pre-Operational Sentinel-3 Snow and Ice Product" (Kokhanovsky et al., 2019). Shortly, the albedo is retrieved from the
calibrated, and geo-located top of atmosphere radiances measured by OLCI at 865 nm and 1020 nm and the snow grain size
is computed from albedo using the ART theory (Kokhanovsky and Zege, 2004) as in our case, but with a largely different
algorithm. Surface snow is assumed to be clean (which is often the case in Antarctica, Warren et al., 2006), vertically and
horizontally homogeneous over a flat, 300 m pixel.



The $d_{opt}$ measured by Multiband (MB) and retrieved with SICE for D5 and D17 (for 140 and 150 clear-sky dates according to the CSI) are shown in Figure 6. At both locations, a group of SICE $d_{opt}$ has very low values independently of the value of corresponding MB $d_{opt}$, likely because of the presence of undetected clouds on the pixel during the satellite overpass. A statistical test is thus applied to identify outliers between low SICE-$d_{opt}$ values. If the latter are <0.2 mm and that the difference with the MB-$d_{opt}$ exceeds 3 times the average difference, the measurement is considered an outlier and rejected. 29 and 28

outliers (in yellow) were identified for D5 and D17 respectively with 27 common dates among them. The remaining pairs of $d_{opt}$ (in blue) have a Pearson's correlation coefficient of 0.67 for D5 and 0.76 for D17. The RMSE between the MB and OLCI dataset are 0.36 mm and 0.16 mm for D5 and D17 respectively. Still, we believe the RMSE estimate at D17 to be more realistic than the one at D5. Indeed, the 300 m pixel of SLSTR including D5 may not be completely representative of the snow reflectance because of the recurrent presence of blue ice areas and containers, as well as the strong average North-South slope

of the area covered by the pixel (8.3%), that is almost double the average pixel slope in D17 (4.3%). Thus, we conclude that Multiband measurements reproduce well the SICE product with an overall error of 0.16 mm RMS that can be considered as an upper bound, since both the satellite and in-situ measurements are subject to independent errors.

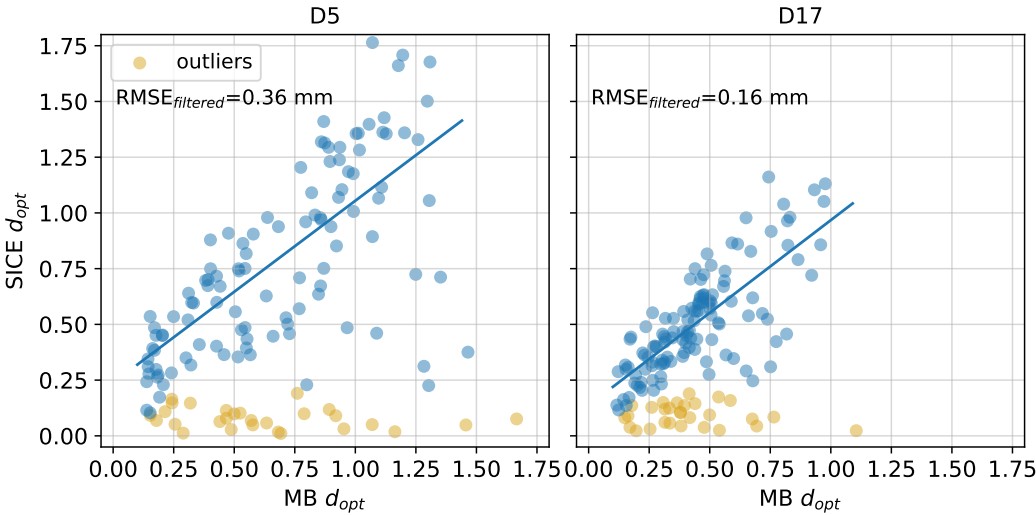

**Figure 6.** Scatterplot of the $d_{opt}$ computed with SICE using OLCI images and measured by Multiband at D5 and D17.

A CNR4 radiometer providing shortwave broadband albedo is installed at D17. To test the quality of our measurements and the performance of our algorithm, we recalculate the broadband albedo from the retrieved $d_{opt}$ and compare to the one

observed by the CNR4. To this scope, we compute the diffuse and direct albedo between 0.28 µm and 2.5 µm with a step of 0.01 µm using equations 2 and 3. One broadband albedo is computed every half-hour, as CNR4 measurements are available every 30 minutes. The spectrum shape and the direct to diffuse ratio of the incoming shortwave radiation are computed with SBDART for a pristine, sub-arctic winter atmosphere. Simulations were made for solar zenith angles between 43° and 70° and cloud optical thickness of 0 and 30 for clear sky and overcast conditions respectively. The broadband albedo is computed as





follows:

$$\alpha^{BB} = \frac{\int_\lambda I^{dir}(\lambda)\alpha^{dir}(\lambda)d\lambda + \int_\lambda I^{diff}(\lambda)\alpha^{diff}(\lambda)d\lambda}{\int_\lambda I^{dir+diff}(\lambda)d\lambda} \tag{4}$$

The resulting broadband albedo measured at solar noon by Multiband and by the CNR4 are shown in Figure 7 (solid line). The 10-90th percentile of the broadband albedo observed during the entire day is also shown (shaded area). The intraday variability of the albedo measured by the CNR4 is much wider, as its mean standard error is of 0.061 for the CNR4 measurements and of
0.008 for Multiband's. This difference, as well as the presence of non-physical albedo values above 1, is explained by sensor tilt and by the presence of slope in the footprint of the CNR4 sensor due to the recurrent presence of sastrugi at D17 and to the relatively low installation of this sensor (<1 m). Sastrugi being transitory bedforms, this slope is dynamic and it is not possible to correct the broadband albedo reliably from its effect, as opposed to spectral albedo (Picard et al., 2020). Still, it is possible to reduce the effect of the local slope by considering only the broadband albedo measured at noon (solid lines in Fig.
7), that is the least affected by tilt and slope as the sun is at its highest. At noon the two timeseries compare favorably, with a Pearson correlation coefficient r=0.51, a mean negative bias of 0.009 of the CNR4 and standard error of 0.060. Significant differences above 0.1 of the two albedo values are 6% of measurements and may be explained by the presence of persistent sastrugi structures or by a tilt of the sensor. Indeed, if we exclude days with CNR4 measurements of the broadband albedo >1 (75% of all days), potentially due to the presence of strong slopes and clear sky conditions, the correlation of the timeseries of
broadband albedo measured at noon increases, with r=0.62, a mean negative bias of 0.013 of the CNR4 and standard error of 0.051.

## 3 Results

### 3.1 SSA and $d_{opt}$ timeseries

The timeseries of retrieved SSA and $d_{opt}$ are shown in Figure 8. Even though these two variables are strictly inversely related,
the visual interpretation of $d_{opt}$ is more suitable for large grains, and conversely, the SSA is more suitable for small grains, the reason why both are shown. In the following, we mainly refer to $d_{opt}$ variations as the prevailing conditions in summer in our study area (melt and wet metamorphism) favor large grains.

The retrieved $d_{opt}$ vary between 0.11 and 1.7 mm at both stations, with mean values of 0.55 and 0.43 mm for D5 and D17 respectively. The variations at both locations are correlated, with a Pearson's correlation coefficient between the two timeseries
of $r$=0.72. Still, the $d_{opt}$ at D5 is larger than at D17 for 70% of all days with valid measurements. The variations of $d_{opt}$ occur at the seasonal scale but also at more rapid scales. On the seasonal scale, the mean $d_{opt}$ is lowest in October (0.20 mm at both locations) and highest in January (0.82 mm at D5, 0.54 mm at D17). Super-imposed on these seasonal variations, we observe occasional sharp decreases of $d_{opt}$ followed by a rapid increase with typical timescales of 1–10 days. These rapid variations usually co-occur at both locations, which confirms the geophysical nature of these changes, i.e. excluding
measurement artifacts. Given this high correlation, we mostly interpret the timeseries at D17 in the following.



**Figure 7.** Comparison of the broadband albedo computed from the retrieved $d_{opt}$ (red) and observed by the CNR4 sensor at D17 (blue) at solar noon (solid line). The 10[th] and 90[th] of the daily albedo measurements is also shown (shaded area).

## 3.2 $d_{opt}$ variations and meteorological conditions

The timeseries of $d_{opt}$ measured at D17 are here compared in Figure 9 with daily snowfall amount (SF), snow drift (SD), daily maximum surface temperature ($T_{s,max}$), and average surface height (SH). Snowfall is taken from the ERA5 reanalysis (Hersbach et al. (2018)). The ERA5 grid point corresponding to D17 is situated at 66.75°S, 139.75°E, 3.5 km from D17,



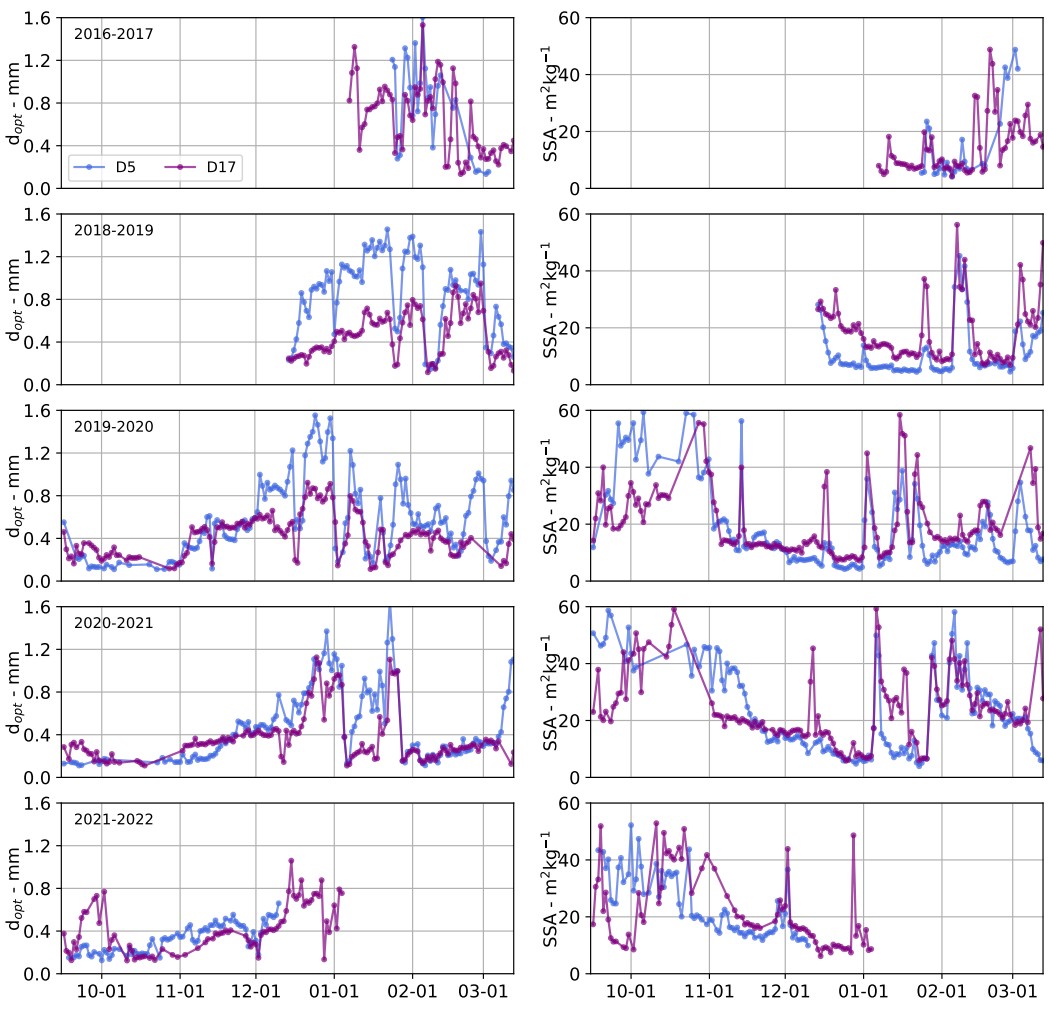

**Figure 8.** Timeseries of $d_{opt}$ (left) and SSA (right) retrieved from Multiband acquisitions in D5 and D17.

at an altitude of 430 m a.s.l. (versus 415 m a.s.l. for D17). Only snowfall >0.5 $\mathrm{kg\,m^{-2}d^{-1}}$ are taken into account to avoid considering spatially confined precipitation that the reanalysis spreads over the whole pixel. The daily amount of drifting snow is measured in-situ by the FlowCapt[TM] instrument (Chritin et al., 1999), that estimates the drifting snow flow from the sound generated by the impacts of the snow grains on the instrument. Although it reproduces well the occurrence of drifting snow events, this instrument underestimates the actual drifting snow flow (Trouvilliez et al., 2014). Moreover, FlowCapt is mounted

approximately 1 m above the ground and misses events of snow mobilization below a 1 m height. The maximum daily surface temperature $T_{s,max}$ is retrieved from the timeseries of the incoming and outgoing longwave irradiance measured and averaged every 30 minutes by the CNR4 on the AWS at D17, using the Stefan-Boltzmann law and a surface emissivity of 0.98. The daily average surface height is retrieved from SR50 distance measurements with respect to a surface located 2.25 m below the sensor





for the first year of measurements (2016-2017) and 1.75 m below for the following. Days with rain events are distinguished by

the conditions $SF > 0$, $T_{s,max} > 0$ and $T_{air,max} > 0$.

Snowfall is reported for 34% of the measurements. The percentage drops to 28% if only days with SF>1.0 $\mathrm{kgm^{-2}d^{-1}}$ are considered. December and October have, on average, the minimum and maximum number of days of snowfall per month, 19% and 47% respectively. In terms of intensity, the month with minimal snowfall is still December, with a daily average of 0.50 $\mathrm{kgm^{-2}d^{-1}}$ over the years of measurement and a limited inter-annual variability (0.32 $\mathrm{kgm^{-2}d^{-1}}$ in 2021 to

0.64 $\mathrm{kgm^{-2}d^{-1}}$ in 2020). February is the snowiest month, with a daily average of 3.12 $\mathrm{kgm^{-2}d^{-1}}$ but a larger year to year variability (0.93 $\mathrm{kgm^{-2}d^{-1}}$ in 2020 to 6.28 $\mathrm{kgm^{-2}d^{-1}}$ in 2021).

Snow drift is almost ever present (90.6% of measurement days) according to the FlowCapt[TM] measurements. The few periods without snow drift last between 1 and 3 days. The median daily amount of snow transported by the wind is 510 $\mathrm{kgm^{-2}d^{-1}}$, while the maximum exceeds 24 $\mathrm{Mgm^{-2}d^{-1}}$. Its intensity is strongly variable among seasons and years. The months with

the weakest and strongest mean daily snow drift (154 $\mathrm{kgm^{-2}d^{-1}}$ and 5388 $\mathrm{kgm^{-2}d^{-1}}$) are January 2017 and September 2021. On average, December and January have the weakest mean daily snow drift (1748 $\mathrm{kgm^{-2}d^{-1}}$), October and November are intermediate (1930 $\mathrm{kgm^{-2}d^{-1}}$), September, February and March the strongest (3068 $\mathrm{kgm^{-2}d^{-1}}$). Still, the inter-annual variability for a same period is sometimes stronger than this inter-seasonal variability. For example, the daily mean for January 2021 is 323 $\mathrm{kgm^{-2}d^{-1}}$, and over 10 times more for January 2020 with 3389 $\mathrm{kgm^{-2}d^{-1}}$.

The timeseries of the two complete years (2019/2020 and 2020/2021) show that the maximum daily surface temperature $T_{s,max}$ is, on average, lowest in September (-17.2°C) and highest in December (-0.2°C). A daily maximum surface temperature $T_{s,max} \geq 0$ °C is hereinafter used as a qualitative indicator of surface melt. Days satisfying this criterion are 3% in November, 48%in December, 27% in January and 7% in February, none during the other months.

The surface height is measured by two sensors from 2017 to March 2021 and only one during the season 2021/2022.

The variations measured by the two sensors are similar and correlated (r=0.93). The surface height decreases on 59% of days of measurements with respect to the previous day, with an average decrease of 14.5 mm. During the remaining 41% of measurement days, the surface height increases of 19.5 mm on average. These timeseries are characterized by occasional, sharp increases followed by long, slow decreases throughout all seasons.

### 3.2.1    Grain size variations during snow drift

We investigate the dynamics of $d_{opt}$ for days with snow drift (SD>0) and neither significant snowfall or surface melt occurring. This is as much as 48.5% of measurement days. Figure 10a shows the distribution and recurrence of the $d_{opt}$ values measured during days when these conditions are met. The $d_{opt}$ values span between 0.11 mm and 1.53 mm, with an average of 0.43 mm. Figure 10b shows the distribution of the variations of the snow surface height between one day and the former. Most changes of the snow height are close to 0. Of the observed $\Delta$SH, 67% are negative, with a median value of -3.3 mm. Therefore, on average,

the snow drift at D17 causes the erosion of the snow surface. The remaining 33% of positive $\Delta$SH range between +0.1 mm and +186 mm. Their 50[th] and 90[th] percentiles are +3.0 mm and +20.0 mm respectively. Higher $\Delta$SH are observed occasionally (7 days). Days of accumulation ($\Delta$SH>0) also have, on average, lower daily mean wind speed and lower daily drifting snow





**Figure 9.** Timeseries of d$_{opt}$ retrieved from Multiband acquisitions at D17 (purple), cumulative daily snowfall (SF, dark blue) cumulative daily snow transport (SD, green), maximum daily surface temperature ($T_{s,max}$, dark green) and daily average surface height measured by two SR50 sensors during the first four seasons and only one during the latter (SH, light blue). Days in which rain occurred are marked by orange diamonds. Values of $T_{s,max} \geq 0$ are outlined by bigger points. The shaded gray area marks d$_{opt} > 0.64$ mm which is the threshold proposed by Vandecrux et al. (2022) to classify melt conditions.





amount than days with surface erosion ($\Delta$SH<0, 7.9 ms$^{-1}$, 1500 kgm$^{-2}$d$^{-1}$ against 8.9 ms$^{-1}$, 2500 kgm$^{-2}$d$^{-1}$). Figure 10c shows the distribution of $d_{opt}$ variations between the day of measurement and the preceding day. The majority of the $\Delta d_{opt}$

are close to 0, with a mean daily grain growth of 0.01 mm. The $\Delta d_{opt}$ is positive for 60% of days and negative for 40% of days. It ranges between -0.84 mm and +0.67 mm, although 83% of measurement span a narrow range between -0.1 mm and +0.1 mm, which is approximately the measurement uncertainty.

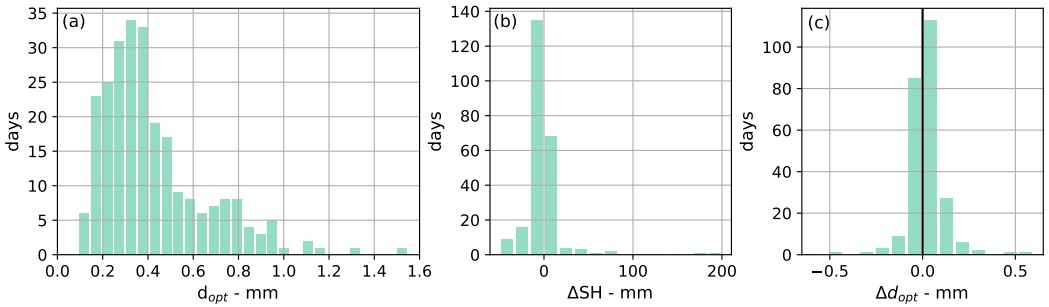

**Figure 10.** Distribution and recurrence of the $d_{opt}$ values measured in days with snow drift but no significant snowfall or surface melt (a). Distribution of the snow height variations (b) and the $d_{opt}$ variations (c) before and after a day with snow drift.

Figure 11 highlights four short periods with active snow drift but limited or absent snowfall and surface melt. Most of the remaining days are isolated and therefore are not suitable to depict the evolution of the snow grain size in the sole presence of

snow drift. In Figure 11a, $d_{opt}$ slowly increases in the presence of wind drift and in the absence of snowfall. Figure 11b shows a similar evolution of $d_{opt}$ but in the presence of weak snowfall that does not interfere with the grain size evolution. As the grain size during both these periods is surprisingly stable, we compare our observations with the estimate of the $d_{opt}$ growth due to metamorphism alone modeled following Carmagnola et al. (2013) (dotted blue line). For both, the grain growth predicted by the model is slightly faster than the measured one. Figure 11c and 11d show a decrease of the $d_{opt}$ during a 2–4 day period

under the sole action of wind drift. Four similar episodes of decreasing $d_{opt}$ lasting 2-3 days were identified throughout the complete dataset. The snow height, during these three first periods, decreases at an average rate of -5.39 mm day$^{-1}$, in line with the mean $\Delta$SH of -5.22 mm day$^{-1}$ observed for all the days with only snow drift. However, the snow height measured during this short timeseries occasionally increases, clearly showing the deposition of drifting snow.

### 3.2.2   Grain size variations during snowfall

The measurements of $d_{opt}$ acquired on days preceded by snowfall (at least 0.5 kgm$^{-2}$d$^{-1}$) are here considered (34.0% of all days). Surface melt and snow drift are detected on 9.0% and 98.3% of these days respectively. Snowfall usually brings small grains onto the surface. Domine et al. (2007) observed freshly precipitated snow with $d_{opt}$ values up to $d_{SF}$=0.2 mm. This value is thus taken as a high threshold for the detection of snowfall here. The distribution of the measured $d_{opt}$ values is shown in Figure 12a. Only 22.6% of measurements satisfy $d_{opt}<d_{SF}$, only 2.8% if we consider days with simultaneous melt. Figure

12b shows the distribution of the variations of the grain size $\Delta d_{opt}$ before and after snowfall. It is divided into measurements



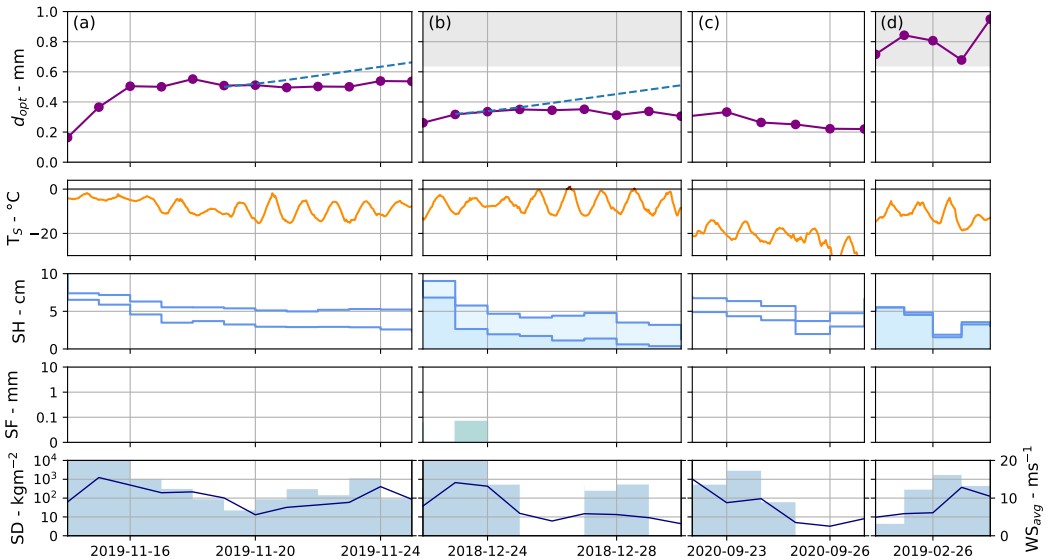

**Figure 11.** Four examples of timeseries of $d_{opt}$ and meteorological variables observed during periods with snow drift and no or negligible snowfall and surface melt. Shown are the surface temperature ($T_S$), the snow height (SH), the snowfall (SF), snow drift (SD) and wind speed ($WS_{avg}$) intensities. Episodes of snow drift and $d_{opt}$ increase in the absence of melt and snowfall (a) and in presence of light snowfall (b). Two episodes of snow drift and $d_{opt}$ decrease in the absence of snowfall and melt (c,d).

with positive changes of the snow height $\Delta d_{opt}$ (54.8%, in green) and negative ones (45.2%, in orange). Both positive and negative $\Delta d_{opt}$ are present. After snowfall, $d_{opt}$ decreases in 53.1% of cases, with a median decrease of $d_{opt}$ of -0.09 mm. In the remaining 46.9% of events, the median increase is +0.05 mm. Considering only the most intense snowfalls ($>10$ kgm$^{-2}$), on average, the surface height increase and $d_{opt}$ decrease account for 77.8% and 70.4% of measurements respectively. Figure 365 12c shows the snow height variations as a function of the intensity of the daily average wind speed during snowfall, the mean of all events being 9.9 ms$^{-1}$. The two variables have a weak, negative correlation ($r$=-0.22).

### 3.2.3 Grain size variations during melt

We analyze the $d_{opt}$ evolution with surface melt, determined as $T_{s,max}>0$. Melt occurs during 17.7% of measurement days, but with a large seasonal spread. The observations from Domine et al. (2007) and Gallet et al. (2014) show that wet snow and melt-370 freeze crusts span a large range of $d_{opt}$ values, from 0.3 mm to above 1 mm. For this reason at least, it is not possible to define a precise threshold for $d_{opt}$ to assess the presence of liquid water in surface snow. Nevertheless, Vandecrux et al. (2022) found that the value of 0.64 mm well represents the limit grain size between days of no or negligible snow melt ($\leq 1$ kgm$^{-2}$d$^{-1}$) and days of intense snow melt ($>1$ kgm$^{-2}$d$^{-1}$) over the Greenland ice sheet. This value, hereinafter called $d_{melt}$, is employed as a threshold to discuss the $d_{opt}$ evolution in the presence of surface melt according to the presence of snowfall and wind drift.





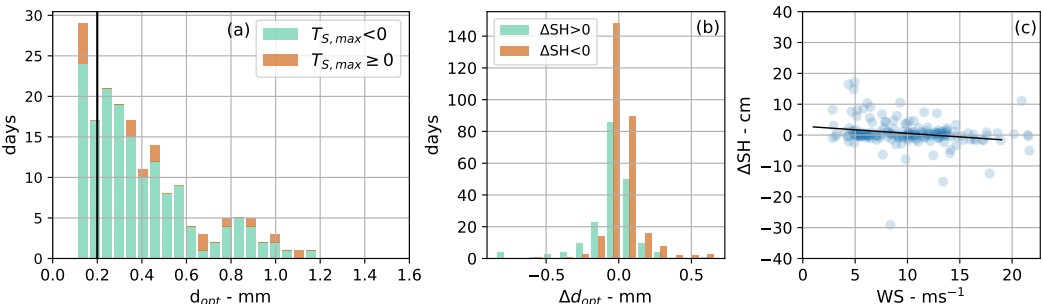

**Figure 12.** Distribution of the $d_{opt}$ values measured in days following snowfall of at least 0.5 $\mathrm{kgm^{-2}d^{-1}}$ (a), snow height variation (n=177 events of at least 0.5 $\mathrm{kgm^{-2}d^{-1}}$) as a function of the snowfall (SF) intensity (b) and as a function of the average daily wind speed (WS) during snowfall (c).

We compare the occurrence of melt during days with grains larger than $d_{melt}$. Overall, $d_{opt} > d_{\mathrm{melt}} = 0.64$ mm occurs on 21.0% of the days, but only 35.8% of them are concomitant with surface melt. These days are rarely isolated (5.6%) but rather grouped into 13 short periods of high $d_{opt}$ lasting 3 to 15 days. These periods were classified according to one of three types of $d_{opt}$ evolution and interaction with snowfall and snow drift that are represented in Figure 13. In Figure 13a, $d_{opt}$ increases above $d_{melt}$ during a day with $T_{s,max} > 0$, is stable during the three following days characterized by the absence of snowfall

and wind drift. Then, $d_{opt}$ starts to decrease four days later after a consistent snowfall (5 cm increase in snow height) but still in the absence of snow drift. Figure 13b shows a similar evolution but with some snowfall and snow drift during the event. At the beginning, the $d_{opt}$ increases above $d_{melt}$ and settles for around ten days, with only a minor snowfall (2019-12-20) that does not affect $d_{opt}$. On the 2019-12-29, a consistent snowfall that brings over 30 $\mathrm{kgm^{-2}d^{-1}}$ over four days begins, concomitantly with strong snow drift. In this context, $d_{opt}$ decreases below $d_{melt}$ only during the fourth day of snowfall, while

precipitation is still intense (>5 $\mathrm{kgm^{-2}d^{-1}}$) but the intensity of snow drift is reduced by two orders of magnitude with respect to previous days (2020/01/02, <100 $\mathrm{kgm^{-2}d^{-1}}$). In Figure 13c $d_{opt}$ is above $d_{melt}$, then decreases due to a snowfall and settles until the middle of the period. In the following days, the condition $T_{s,max} > 0$ is never satisfied, meaning that surface melt does not occur. Nevertheless, starting on 2019-01-26, $d_{opt}$ increases again, over $d_{melt}$, during a few days of intense snow-drift (>$10^4$ $\mathrm{kgm^{-2}d^{-1}}$) and the snow height decrease. The period ends with a snowfall in the presence of lighter snow drift

(2019/02/04, 100 to 1000 $\mathrm{kgm^{-2}d^{-1}}$). Finally, during all the periods considered in Fig. 13, as well as the ones in Fig. 11, the threshold wind speed for the detection of snow drift at a 1 m height is approximately 5 $\mathrm{ms^{-1}}$, independently from the grain size.

     Figure 14a shows the distribution of the measured $d_{opt}$ during days with surface melt. Only 42.4% of $d_{opt}$ measured during these days exceed $d_{melt}$, however, the average $d_{opt}$ (0.56 mm) is 0.14 mm higher than the average $d_{opt}$ on days without melt,

confirming its role in the grain growth. During melt events, $d_{opt} > d_{melt}$ is met equally for days with (66.3%) and without (33.7%) snow drift (34.4% and 35.7% respectively). Figures 14b and 14c show the distribution of the surface height variations $\Delta SH$ and $d_{opt}$ variations $\Delta d_{opt}$ between days with actual melt and the preceding day. The distribution of $\Delta SH$ ranges

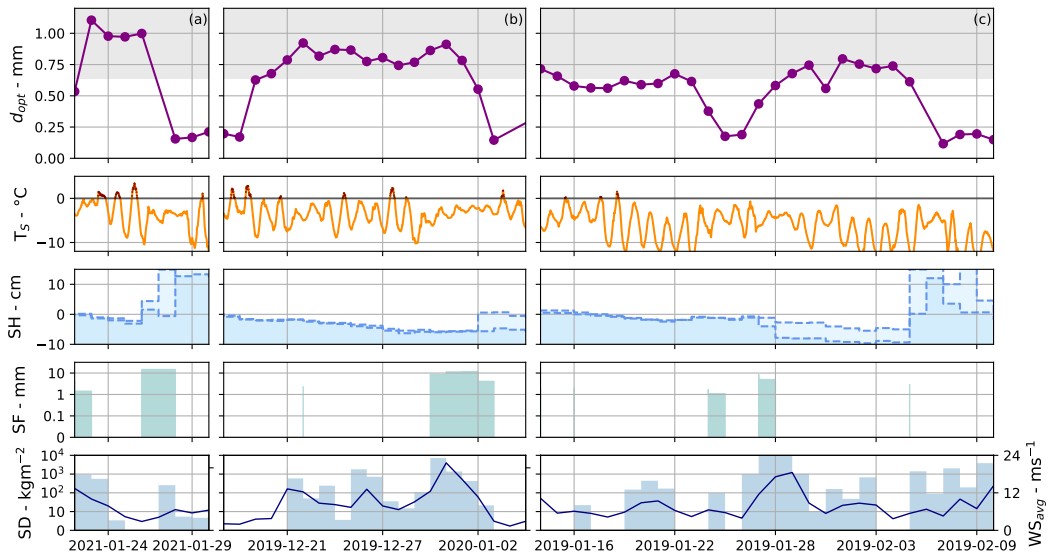

**Figure 13.** Three examples of timeseries of $d_{opt}$ and meteorological parameters observed during periods with large grains ($d_{opt} \geq 0.64$ mm). Shown are the surface temperature ($T_S$), the snow height (SH), the snowfall (SF), snow drift (SD) and wind speed ($WS_{avg}$) intensities. An episode of surface melt during a period of limited snowfall and snow drift (a). A melt event that concurred with snowfall and snow drift (b). An episode o of erosion of the snow surface to a layer with $d_{opt}$ values of a previous melt event, without surface melt (c).

between -21 mm and +164 mm, but despite the large positive upper bound, the median is negative, -2.9 mm. In the same line, 56.6% of the events resulted in a decrease of the surface height, which is expected due to the compaction of the snowpack

associated to melt. The few melt events that concurred with snowfall (16 events, 3% of total measurements) are characterized by a similar range of $\Delta SH$, but with a higher median (+0.1 mm) than those without snowfall (-3.5 mm). The median $\Delta d_{opt}$ is an increase of 0.01 mm. Besides, the distribution of $\Delta d_{opt}$ for the days with and without concurring snowfall are slightly shifted. $\Delta d_{opt}$ is higher without snowfall (+0.018 mm) than with snowfall (-0.121 mm) and with snow drift (+0.013 mm) than without (+0.001 mm).

# 4   Discussion

## 4.1   Multiband

In the conception of the Multiband albedometer, multi-band acquisitions were chosen over spectral ones in order to reduce its energy consumption. As a result, Multiband could be deployed in multiple remote locations of the Antarctic continent, where it has been acquiring reliable measurements in complete autonomy. Although only six bands are exploited for the processing and

only three are used for the grain size retrieval, Multiband observations have compared favorably with satellite retrieved grain size and in-situ measurements of the broadband albedo. Moreover, the coherence between the grain size timeseries measured





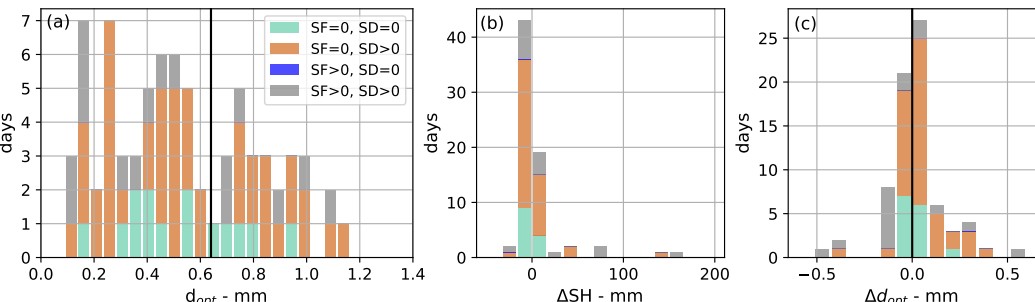

**Figure 14.** Distribution and recurrence of the $d_{opt}$ values measured in days with energy available for surface melt with respect to the occurrence of snowfall and snow drift (a). Distribution of the snow height variation (b) and the $d_{opt}$ variation (c) before and after a day of melt with respect to the occurrence of snowfall or snow drift.

at D5 and D17 is remarkable. The two locations, despite an altitude difference of approximately 300 m, have similar grain size values during fall and at the end of the summer season (0.33 mm at D5, 0.31 mm at D17) and clearly diverge only during the melting period in full summer (0.74 mm at D5, 0.52 mm at D17). In spite of the heterogeneity of the two locations as 415 seen from the field, the similarity of the grain size timeseries provides confidence to the Multiband measurements. Moreover, it justifies the use of simulations and reanalysis at multi-kilometer resolution in this area (Hersbach et al., 2018; Amory et al., 2021).

## 4.2 Main processes driving the snow grain size

Our observations depict the evolution of the snow surface grain size in the windy coastal locations of D5 and D17 in Adelie 420 land, East Antarctica, over 5 summer seasons, for a total of nearly 600 days with successful retrieval. Through the comparison with the timeseries of the meteorological variables and snow height observed at D17, we relate the complex evolution to the multiple competing environmental drivers of the grain size, namely dry metamorphism, snow drift, snowfall and melt. Overall, our results show a seasonal cycle characterized by small grains in spring and autumn and larger grains in summer. These variations directly impact the broadband albedo with typical values between 0.82 and 0.75 (Fig. 7), implying a large variation 425 of the short wave net radiation absorbed in the snowpack. However, our results also suggest that the evolution of the surface albedo in the presence of snow drift due to strong wind is less predictable than it would be in the absence of these drivers.

Snow drift at D17 has mainly an erosive character, as observed during 67% of measurement days when snow drift occurred without snowfall nor melt. Increases of the surface height are less frequent (33%) but significant in terms of magnitude, with a median and maximum snow height change of +3 mm and +186 mm respectively. However, these large variations of the snow 430 height do not coincide to major changes in the grain size. Indeed, most grain size variations (83%) range between -0.1 mm and +0.1 mm. Both episodes of snow accumulation and decrease of the grain size during (58% of days with snow drift) imply the deposition of small, drifting grains on the snow surface, as suggested by Domine et al. (2009) after the observation of similar dynamics in the Canadian Arctic. Still, even when the grain size increases from one day to another, growth happens at a very



slow pace (median +0.03 mm). This noteworthy stability of the grain size challenges the expectation of growing snow grains
under the action of dry metamorphism during the summer season. We think that this limited growth may also be related to
the input of small grains deposited during snow drift, despite the overall erosion of the snow surface in this period. For the 28
measurements with increase of grain size over +0.1 mm, the average grain size was large (0.63 mm) even though the average
surface temperature during those measurements remained below -10°C. Such large grains without melt are explained by the
erosion of the snow surface, likely occurred until a layer of very cohesive, coarse-grained snow, as in Figure 13c.

The interaction of snowfall and wind drift also has an important impact on the observed snow grain size. Indeed, days with
snowfall frequently featured surface erosion (45%) and an average grain size increase of +0.02 mm, against the average grain
size decrease of -0.03 mm of days with accumulation (55%). The erosion of the snow surface reveals that precipitation is
completely swept away by the wind for almost half of the snowfall events at this location, at least for the spots under the
surface height sensors. Moreover, the evolution of the snow grain size after episodes of erosion during snowfall follows similar
dynamics to those of erosion alone, with 56% of grain size increase and 44% of decrease (60% and 40% in the case of erosive
snow drift alone). Jakobs et al. (2021) mention frequent snowfall as a determinant factor to limit the length of low albedo
periods leading to surface melt. In windy areas, however, the frequency of snowfall leading to actual accumulation should be
considered, as almost a third of all snowfall events result in surface erosion followed by an increase of the snow grain size.
The result of these complex interactions in D17 is that the commonly expected outcome of snowfall– an increase of the surface
height and a decrease of the grain size– is only met on 33% of days with snowfall, and the average grain size change during
precipitation is in fact negligible (<0.01 mm). Finally, the weak, negative correlation between the daily snow accumulation and
the wind speed (r=-0.22) suggests that accumulation is more likely for more intense snowfall and lower wind speeds and lets
us mark an average daily wind speed of 12 $\mathrm{ms^{-1}}$ as a rough boundary between accumulation and erosion at D17.

The evolution of the grain size in the presence of surface melt is also complex. The expected output for surface melt, which
is an increase of the grain size and a moderate decrease of the surface height (due to the compaction of the snowpack), is
only met on 27% of days with melt. Indeed, 37% of melt days saw an increment of the snow height and an average grain size
decrease. However, even within days with surface erosion (63%), the sharp increases of the grain size that would be expected
because of surface melt are occasional, while the average increase is +0.05 mm only. We believe this to be due to the input
small grains at the surface that contrast or limit the melt-driven grain growth. As a consequence of these interactions, the $d_{opt}$
is, on average, rather stable in days with surface melt, with a mean grain growth of 0.02 $\mathrm{mm\ d^{-1}}$ only.

Overall, our main conclusion is that the prevailing dynamics caused by snow drift at D17 is to prevent sharp changes of
the snow grain size. Indeed, snow drift 1) delivers small grains onto the surface, which contrasts the action of metamorphism,
either dry or wet, and 2) prevents the deposition of snowfall. The major drivers of the grain size increase – metamorphism –
and decrease – precipitation – are thus inhibited, granting a relative stability of the grain size with respect to the ensemble of
snowfall and surface melt events. Nonetheless, rare, sharp increases of the $d_{opt}$ caused by the wind were observed. They are
mainly due to the erosion of the surface snow until a refrozen melt layer. It is possible that the deposition of small grains over
these layers is prevented by the strong cohesion of the surface, as suggested by Comola et al. (2019).





## 4.3 Limitations

Some limitations of this study should be mentioned. On the basis of the analysis carried out in Section 2.5 and 2.6, we estab-
lished an upper-bound RMSE=0.16 mm for the $d_{opt}$ retrieved from Multiband measurements. Although this value is higher
than most variations of the grain size measured from one day to another, statistical considerations remain valid. Moreover,
most measurements are likely to have a lower RMSE as the satellite-retrieved $d_{opt}$ and in-situ measurements have independent
error sources. In addition, during the slope correction, the evolution of the observed albedo during the day is considered due
to the surface slope only. This hypothesis is sometimes incongruous in summer due to frequent high surface temperatures and
occasional surface melt, that cause a rapid evolution of the surface properties of snow. Still, we consider that the effect of this
assumption of the grain size timeseries is not crucial. Indeed, 53% of the days of measurement were classified as overcast,
which means that the effect of slope is negligible during these days, and the slope correction in these cases returns an average
of the measured albedo, thus a daily mean albedo. For the days classified as sunny, the average daily surface temperature is
-10.2°C, which implies a generally slow rate of metamorphism. Moreover, during clear-sky days with higher surface temper-
atures, eventual major variations of the surface albedo during the day would be perceived as a strong slope by the model or
have a high RMSE, and would therefore be rejected. Another limitation is the combination of data with different spatial rep-
resentativeness, either because they have a small, non-overlapping footprint (albedometer, surface height sensors, CNR4) or a
large scale (OLCI, ERA5 reanalysis) relative to the local heterogeneity of the area. This necessarily leads to an increase of the
observed "unexpected" situations. Finally, the FlowCapt[TM] instrument can not distinguish the flow of drifting snow advected
from the surface and precipitating particles. Indeed, snow drift is detected during 87% of days without snowfall and almost
always during days with snowfall (98%). However, the mean daily wind speed during days with snowfall is 9.9±4.2 ms$^{-1}$,
which is enough to advect eventually settled freshly fallen snow (JDoorschot et al., 2004; Clifton et al., 2006).

## 5    Conclusions

The spectral albedo of surface snow at D5 and D17, two windy locations in Adélie Land, East Antarctica, was measured over
a small area (a few square meters) during several years. To acquire this unique dataset, a specifically designed autonomous
spectral radiometer (named Multiband) was developed and was presented here. Instrumental and environmental artifacts were
corrected to retrieve timeseries of accurate effective optical diameter of snow for five summer seasons. The analysis of the
uncertainties, the comparison with satellite-retrieved snow grain size and with in-situ broadband albedo measurements, as well
as the coherency between the grain size observed at D5 and D17 allow us to conclude that Multiband is a reliable instrument for
the retrieval of the daily grain size and the snow albedo. The timeseries from D17 were then compared to snowfall, snow melt,
snow drift and other meteorological variables. The results show that the interaction of these processes cause a complex evolution
of the snow grain size, and thus of the surface albedo. Overall, our results show that the net effect of wind and snow drift is a
relative stability of the grain size of the snow at D17. Indeed, snow drift often deposits small grains onto the surface contrasting
metamorphism, but it also prevents the renewal of the surface snow as it swipes away precipitation. Exceptionally, however,
the wind causes sharp increases of the grain size due to the uncovering of cohesive, old snow layers over which the deposition



of drifting snow is ineffective. The wind-driven evolution of the grain size of the superficial snow layer at D17 is therefore significantly different from what it would be in the presence of dry metamorphism, snow melt and snowfall alone. Accurate modeling of this evolution is thus necessary in order to better constrain the present and future changes of the surface albedo and the strength of the albedo-related feedbacks in coastal Antarctica.

*Code and data availability.* Spectral albedo and grain size timeseries and codes to generate the figures will be made available in an online repository upon acceptance of the paper for publication.

*Author contributions.* SA, GP and LA designed the study. LA and GP designed, assembled and installed the Multiband instrument. GP, LA and VF collected data in the field. SA implemented the processing of the observations, performed the analysis and wrote the manuscript. GP, LA and VF discussed and revised the manuscript.

*Competing interests.* The authors declare that they have no conflict of interest.

*Acknowledgements.* This study has been supported by the Agence Nationale de la Recherche projects 14-CE01-0001 ASUMA and 1-JS56-005-01 MONISNOW, the Centre National d'Etudes Spatiales project Trishna, the Insitut Polaire Emile Victor project 1110 NIVO, the European Space Agency project 4D Antarctica, the GLACIOCLIM project SAMBA. The authors acknowledge Eric Lefebvre for the development of the electronics for the Multiband instrument, Charles Amory for his contribution in the exploitation of the FlowCapt mea-
surements and Baptiste Vandecrux for his contribution in the comparison of in-situ measurements with SICE and for his valuable feedback on the manuscript.



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
