# Peer review of "Dynamics of the snow grain size in a windy coastal area of Antarctica from continuous in-situ spectral albedo measurements"

_The Cryosphere, 2022_

## Author Comment (AC1)

The authors would like to thank the reviewer for their comments and feedback. Our answers are presented below in blue. For technical and specific comments, the check mark indicates that the suggested modification was implemented.

**General comments**

There is a huge need for precise surface albedo measurements, which require a thorough calibration and estimation of the measurement uncertainties. The different processing and calibration steps are described very clearly, however I still have some open questions:

**(1)** I am wondering about the cross calibration of the two sensor heads. Maybe I missed it in the manuscript, but what were the weather conditions during the side-by-side observations? In my opinion, this cross calibration should be performed using a purely diffuse light source only, as otherwise the differences in the cosine response of the two sensor heads are somehow corrected as well, even though the actual cosine response correction is performed in the next step and only for the upward-looking sensor which receives direct radiation. In my eyes, a cross-calibrating during conditions with direct incident radiation would lead to a compensation for effects between the two sensor heads which are not occurring during the actual measurements (as the downward-looking sensor head only receives diffuse radiation). Please correct me if I am wrong, but I would like you to comment on my line of thought.

Using a diffuse light source for the cross-calibration of the albedometer channels would be ideal. The choice of the upward facing position of the collectors was intentional and part of our calibration protocol, in order to collect information on the zenithal variations, but in the end it was not a great idea. Nevertheless, the coefficients used for the cross-calibration are averaged for a wide range of solar zenith angles varying between 79° and 45° for D17 and between 67° and 44° for D5. Still, the relative standard deviation of the coefficients is well below 1% for all wavelengths used for the retrieval of SSA and $d_{opt}$ for both stations. We therefore believe that the partial compensation of the cosine does not affect the cross-calibration significantly.
With respect to this comment, we propose the following addition to Section 4.3 (limitations): "The cross-calibration was performed with the collectors facing upward, meaning that the light source had a direct component during the experiment. Because this may interplay with the cosine correction, we average the calibration coefficients for a wide range of solar zenith angles (79° to 45° for D17, 67° to 44° for D5). Their relative standard deviation is well below 1% for the wavelengths used for the retrieval of SSA and $d_{opt}$ for both stations, which is a low residual error. However, future calibrations should be performed under diffuse radiation, either during cloudy period or with collectors looking downward".

**(2)** The bandpass filters used for the spectral measurements have a certain spectral width (usually characterized by a FWHM - full width at half maximum). Could you please specify the FWHM of the spectral filters used in the instrument, and also discuss the influence of the non-discrete wavelengths on the retrieval of the optical snow grain size using the ART formulas?

The spectral filters have a FWHM of 25 nm (specified in L94 together). To test the sensitivity of the ART theory to the width of the spectral band, we compared the 'discrete' albedo - computed from the ART theory at the wavelengths corresponding to the spectral filters for a given $d_{opt}$ - and the 'non-discrete' albedo, computed as:

$$\alpha_{n-d} = \frac{\int_\lambda (\alpha_{dir} \cdot I_{dir} + \alpha_{diff} \cdot I_{diff}) \cdot f}{\int_\lambda (I_{diff} + I_{dir}) \cdot f} \tag{1}$$

where $\alpha_{dir}$ and $\alpha_{diff}$ are the direct and diffuse albedo components for a given $d_{opt}$, computed at a resolution of 1 nm. $I_{dir}$ and $I_{diff}$ are the direct and diffuse components of the incoming light flux computed with the SBDART model and $f$ is the Gaussian response function of the filter, with a FWHM of 25 nm. $\alpha_{n-d}$ was computed for each of the filters' central wavelengths and for all $d_{opt}$ in our timeseries. For all wavelengths, the average difference between the 'discrete' and the 'non-discrete' albedo is very low, $< 0.002$, while the maximum difference is $< 0.004$. We therefore conclude that the impact of the spectral width of the filters on the $d_{opt}$ retrieval is negligible.
We propose the following addition in Section 2.4 of the manuscript: "The model is applied considering

that the albedo is measured at 800, 925 and 1050 nm precisely, ignoring the spectral width of the filters (25 nm). Indeed, using Equation 2 and 3 in numerical tests, we found maximum differences of albedo to be less than 0.004 when accounting or not for the spectral band width, which is a negligible error. Finally, the effective optical grain diameter is deduced...".

**(3)** Clear sky index (Lines 157-164): Why is the clear sky index (CSI) not applied to each individual measurement, but the entire day is assumed to be clear sky when 75% of the observations had a CSI below 1.25? It seems applying the clear sky/overcast distinction before calculating the daily average would reduce the uncertainty in the retrieval.

**(4)** This applies also to the remark on Line 216: Would it not be possible to use these filtered measurements if one would apply a CSI to each measurement?

We reply to both comments 3) and 4) here. Following this suggestion, we have tried to apply a CSI value (and thus a direct/diffuse ratio) to every single measurement and have found a marginal gain: 6 and 13 less measurement days are discarded for D5 and D17 respectively. The gain being small, the uncertainties of the retrieved SSA/$d_{opt}$ are virtually unchanged. Our explanation is that the days discarded by the slope correction do not correspond to either clear or overcast sky, but to scattered clouds and strongly variable direct/diffuse light ratio. For this reason, measurements taken during these days are discarded even if a CSI per measurement (thus a direct/diffuse light ratio) is assigned.
A more important gain will come from a direct measurement of the direct/diffuse light ratio as a complement to Multiband measurements.
The following is added in Section 4.3 (limitations): "The direct/diffuse light ratio used for cosine and slope correction corresponds to either clear or overcast sky conditions. Both these ratios are inaccurate for days with scattered clouds and strongly variable direct/diffuse light ratio, that are discarded after slope correction. In order to reduce uncertainties and the number of discarded measurements, a valuable future addition to Mutliband is the direct measurement of the direct/diffuse light ratio".
N.B. while investigating for these comments, we noticed a mistake in the percentage of discarded measurement days at Line 216. This is now corrected: 4.4% for D5 and 6.2% for D17.

**(5)** How broken are these clouds that are still considered to be clear sky? Maybe you can give some more details here from Marty and Philipona (2000), if available?

Marty and Philipona (2000) provide an effective way to detect perfectly clear skies but don't discuss their index with respect to type of clouds. In the pictures taken at noon every day at D17, we observed that many days characterized by the presence of cirrus or haze had a CSI slightly $> 1$ but also had a strong direct solar light flux. Thus, the CSI threshold for the clear-sky classification was increased from 1 to 1.25 to classify these situations as clear-sky rather than overcast.
In the manuscript, we change L162 to "...as we consider a thin or partial cloud cover that still lets direct light penetrate to the ground (e.g. cirrus or haze) to be better represented by clear sky conditions than overcast".

**(6)** The computation of the CSI uses temperature and humidity data from the AWS at D17 (which is above 400 m a.s.l.). What is the influence to use the same data and apply it to D5 (below 200 m a.s.l.)? Is this why you chose to apply the same CSI to one full day as there was no separate AWS available at D5? Or are you using reanalysis data for D5? This needs to be clarified within the manuscript.

Unfortunately, we can not quantify the influence of extrapolating the CSI to D5, and the reanalysis are too coarse resolution to address this question. Intuitively, we expect that the extrapolation is suitable for some situations (clear-sky and overcast) and inadequate for others (scattered clouds).
We propose the following addition to the Clear Sky Index paragraph: "The CSI computed for D17 was also used for D5, as we lack in-situ temperature and relative humidity measurements at the station.

This was preferred to using reanalysis such as ERA5 or MERRA, with a too coarse resolution to capture the difference between the two sites".

**(7)** Why are the SSA/$d_{opt}$ only retrieved from the diffuse albedo spectra (e.g. Line 186 and 202)? If you are correcting for the cosine response of the sensor head and use Eq. 2, is it not possible to retrieve SSA/d_opt from the direct albedo measurements?

The output of the slope correction is a daily diffuse albedo, and the estimation of the SSA/$d_{opt}$ directly uses this output. The signal from the sensor at a given time, even after cosine correction, is still affected by the slope (which is unknown). While retrieving SSA/$d_{opt}$ from such measurements has been done in the past (e.g. Picard et al. 2016), it is much less accurate than after correcting the slope.
In the manuscript, "The two unknown parameters, A and SSA are computed by fitting the model to the observed diffuse albedo at 800, 925 and 1050 nm using a least-square minimization" is changed to "The two unknown parameters, A and SSA are computed by fitting the albedo model (Equation 3) to the daily diffuse albedo provided by the slope correction at 800, 925 and 1050 nm using a least-square minimization".

**(8)** At first glance in Fig. 7, it looks like the variations in SW albedo measured by the CNR4 (shaded blue) are largest during clear-sky days (noon line shows the clear-sky symbol). This underlines your statement in the text that the influence of e.g. sastrugi is largest on clear-sky days. I can follow your argument why you are focusing on the noon observations to reduce this error, but wouldn't this error be even lower if you would focus only on the overcast observations for this test? Especially with the different footprints (CNR4 installed at lower height), I think it is reasonable to try and reduce the uncertainties as much as possible in order to make the comparison as fair as possible between the two approaches. Thus, it would be interesting to see the statistics/deviations if you discard the clear-sky observations in the time series for this test.

We agree with the reviewer and therefore propose the following addition: "Similarly, considering only measurements acquired during days with overcast sky – that are not affected by surface slope – the correlation coefficient increases up to r=0.55, with a mean negative bias of the CNR4 of 0.023 and a standard error of 0.047".

**(9)** When comparing the $d_{opt}$ from the surface albedo measurements with the satellite observations: could you please briefly discuss the influence of the different wavelengths used in the ground-based and satellite retrievals? The different wavelengths would lead to slightly different penetration depths of the radiation into the snowpack, thus the two instruments are 'seeing' slightly different layers of the snowpack.

The SICE algorithm for clean snow uses OLCI reflectances from bands Oa17 and Oa21, centered at 865 nm and 1020 nm, while we retrieve the optical grain size from albedo measurements at 800 nm, 925 nm and 1050 nm. The penetration of light into the snowpack at 1020 nm and 1050 nm is nearly the same (using the formulation from Kokhanovsky (2022)), with differences in the e-folding depths typically < 1 mm. These two wavelengths are the most influential in the SSA retrieval. At smaller wavelengths (800 to 925 mm) the e-folding depth difference is also small, few millimeters to 1 cm according to snow density and grain size. We believe this difference to be negligible for the SSA/$d_{opt}$ retrieval, especially when compared to the impact of the footprint difference between Multiband (few m$^2$) and the OLCI (300x300 m$^2$ pixel).
We propose not to modify the manuscript with respect to this comment.

**(10)** The evolution of the snowpack over the 5 seasons presented in this study is discussed in very good detail. However, the manuscript would definitely benefit from putting this impressive data set in perspective to very similar studies at other Antarctic locations. For example, Libois et al. (2015) presented a multi-year study of SSA evolution retrieved from albedo measurements and discuss the influence of drifting snow. Also, Carlsen et al. (2017) showed the temporal evolution of the SSA from surface albedo measurements on the Antarctic plateau and compared it to in situ and – similarly to

this study – optical satellite observations. A more thorough discussion on how the different studies compare would be an important addition to the discussion.

We propose the following extension of Section 4.2 (L461): "The dynamics observed at D17 may compare to locations with similar characteristics in terms of snow type, temperatures, humidity and wind regimes only. However, they show a remarkable range of possible evolution paths of the snow optical properties involving snow metamorphism, snowfall and snow melt in the presence of snow drift, some of which are regularly observed in literature. Libois et al. (2015a) and Carlsen et al. (2017), for example, described the evolution of the snow SSA over one summer season at two locations over the East Antarctic Plateau. On the Plateau, strong winds are less common than on the coast and SSA usually increases during snowfall and slowly decreases afterwards. However, they both observed punctual episodes of sharp SSA decreases after snowfall due to the removal of the fresh snow layer by snow drift. Also, Vandecrux et al. (2022) and Jakobs et al. (2021) observed punctual decreases of the grain size (or equiv. increases of albedo) during dry, warm summer periods at EastGRIP (Central Greenland), and over the King Baudouin ice shelf (Dronning Maud Land, East Antarctica) respectively, likely due to the deposition of smaller snow grains after drift events. Inversely, our main conclusion...".

**Specific comments**

1. Introduction: wavelength dependence of surface albedo - could you give some typical values from literature?
   We do not find the part of the text this comment refers to.

2. L19: I recommend a quick, short definition of surface albedo at the first mentioning.
   The beginning of the Introduction section was modified as follows: "Snow covered areas have a cooling effect on the climate both at local and global scale because of their high albedo, which is the ratio of the reflected irradiance to the total incident solar irradiance (Zhang et al. 2022a)".

3. Figure 1: Grid lines would help, maybe include a photo of sastrugi/snow drift event as panel c?
   ✓

4. Figure 2: Please highlight the components by annotating the photo ✓

5. Section 2.1: One more sentence to the measurement principle of the snow drift volume would help at this point.
   The end of the paragraph was modified as follows: "...and a FlowCapt$^{TM}$ instrument (Chritin et al. 1999), that estimates the drifting snow flow from the sound generated by the impacts of the snow grains on 1-m long vertical pipes".

6. L124: It would be helpful to give a percentage of how many measurements were filtered out by the SZA criterion and the 1% total irradiance criterion.
   The percentage of measurements excluded for the SZA and 1% criteria are already given in the manuscript, in L207.

7. Figure 5: the axis annotations for SSA and d_opt need a unit for both (m2/kg and mm I believe)
   ✓

8. L251: I think 'reproduce' is a bit misleading here, implicating that the satellite measurements would be the ground truth, even though they come with a lot of uncertainties in themselves (as you mention before).
   "Reproduce well" was replaced by "correlate with".

9. L305: T_air,max is mentioned here for the first time, but never explained. What is the difference to T_s,max?
   $T_{air,max}$ is the maximum air temperature recorded at the AWS during the day. The sentence is reformulated in the manuscript with the explicit definition.

10. L312-: It is nice to see all these statistics and to put inter-seasonal and interannual variability in relation to each other. However, it would be good to compare this to literature values of other

snow drift measurements in Antarctica and maybe put these values into context in terms of weak and strong snow drift events.

The following observations were added to the paragraph describing snow drift in Section 3.2:
1. The range of observed snow drift values is coherent with observations by Amory et al. (2015) for January 2011 at D17.
2. The snow drift at D17 is weakest in the full summer months and strongest at the beginning and the end of the summer season as in Amory et al. (2020).
3. Amory et al. (2020) found that major drift events contribute for over 70% of the total transported snow mass at D17 on an yearly scale, which explains the strong variability of the mean daily snow flow for a same month among years.

11. Figure 11d: the $d_{opt}$ values are within the shaded area to classify melting conditions (caption of Figure 9), however the surface temperature is clearly below 0°C so I do not know how helpful the gray shading in Fig. 11 is, especially as these are cases with no melt. You could consider removing this shading to foster clarity. ✓

12. L357: Some further justification is needed why the 0.2 mm measurement of the grain size of freshly precipitated snow by Domine et al. (2007) is used as an upper threshold here. With the given information, this seems quite arbitrary to me.
The sentence "Domine et al. (2007) observed freshly precipitated snow with d opt values up to $d_{SF}$=0.2 mm. This value is thus taken..." is replaced by "Domine et al. (2007) measured the SSA of over 60 samples of freshly fallen snow. Within this dataset, the lowest SSA value measured is 33 $m^2kg^{-1}$, that corresponds to a $d_{opt}$ of 0.2 mm. Thus, this value, hereinafter called $d_{SF}$, is taken...".

**Technical corrections**

1. L11: decrease ✓

2. L27: liquid water content ✓

3. L85: shortwave infrared ✓

4. L94/96: irradiance instead of radiance ✓

5. Table 1 caption: irradiance ✓

6. L110: Fig. 2 ✓

7. L243: omit additional 'that' ✓

8. Figure 7 caption: 10th and 90th percentile ✓

9. L280: This is the reason why both are shown. ✓

10. Figure 9: the x axis should just be the months, in order to correspond to the other panels for the other years (so no specific 2022-10, 2022-11, but only 10, 11, . . . ) ✓

11. L312: ever-present ✓

12. L314: It would be easier for the reader to stick to the unit kg m-2 d-1, and not switch to Mg ✓

13. L322: Days satisfying this criterion correspond to 3% of the total measurements in November, 48% in December . . . ✓

14. L327: increases by ✓

15. L361: I believe you mean delta SF? changed to ΔSH (snow height)

16. L389: and the snow height decreases ✓

17. Figure 13 caption: 'an episode of erosion' ✓

18. L425: shortwave ✓

19. L460: on days ✓

**References**

Amory, Charles, et al. "Comparison between observed and simulated aeolian snow mass fluxes in Adélie Land, East Antarctica." The Cryosphere 9.4 (2015): 1373-1383.

Amory, Charles. "Drifting-snow statistics from multiple-year autonomous measurements in Adélie Land, East Antarctica." The Cryosphere 14.5 (2020): 1713-1725.

Carlsen, Tim, et al. "Comparison of different methods to retrieve optical-equivalent snow grain size in central Antarctica." The Cryosphere 11.6 (2017): 2727-2741.

Jakobs, Constantijn L., et al. "Spatial Variability of the Snowmelt-Albedo Feedback in Antarctica." Journal of Geophysical Research: Earth Surface 126.2 (2021): e2020JF005696.

Kokhanovsky, Alexander A. "Light penetration in snow layers." Journal of Quantitative Spectroscopy and Radiative Transfer 278 (2022): 108040.

Libois, Quentin, et al. "Summertime evolution of snow specific surface area close to the surface on the Antarctic Plateau." The Cryosphere 9.6 (2015): 2383-2398.

Vandecrux, Baptiste, et al. "The determination of the snow optical grain diameter and snowmelt area on the Greenland Ice Sheet using spaceborne optical observations." Remote Sensing 14.4 (2022): 932.

---

## Author Comment (AC2)

The authors would like to thank the reviewer for their comments and feedback. Our answers are presented below in blue. The check mark indicates that the suggested modification was implemented.

After reading this nice analysis on snow grain size behavior adjacent to Dumont D'Urville station, it is clear that I do not have sufficient expertise in this topic area to provide useful scientific input. On page 22, the authors conclude that snow drift impacts at D17 prevent significant changes to snow grain size due to competing effects. That is, there is relative stability to the surface albedo.

Here are some small items:

1. Lines 37-39: This sentence seems to be saying contradictory things. Higher precipitation should probably be associated with higher albedo.
   We propose the following rephrasing of the sentence: "For instance, Picard et al. (2012) observed an albedo higher than average by 0.03 throughout the summer season for years with higher summer precipitation at Dome C, on the Antarctic Plateau."

2. Line 41: Drop "operated". ✓

3. Line 77: Automatic rather than Automated. ✓

4. Line 113: Rephrase to "buried in the snow to provide temperature stability for the electronics".
   We propose the following rephrasing of the sentence: "...are buried in the snow to ensure the stability of their temperature, thus of the measured signal accuracy".

5. Line 211: "obstructing" rather than "obturating" would be better as much more frequent usage and less jarring. ✓

6. Figure 9: Caption says "The shaded gray area marks $d_{opt} > 0.64$ mm". The gray shading along the top of each panel is continuously gray and does not coincide with $d_{opt} > 0.64$ mm in most locations.
   The gray shading in Figure 9 is modified and present only for days with $d_{opt} > 0.64$ mm. The same modification is applied to Figure 13.

7. Line 543: Complete the reference for Colbeck 1973. ✓

---

## Editor Decision (ED1)

2023-04-11
Submission tc-2022-236

**Dynamics of the snow grain size in a windy coastal area of Antarctica from continuous in-situ spectral albedo measurements**

Sara Arioli et al.

Dear Dr. Arioli, thank you for your submission to be considered for publication in The Cryosphere. After reading reviewer's comments, and my own reading of the paper and answers to reviewers, I think main concerns were thoroughly addressed by the authors. The provided response are clear and modifications to the paper do improve the overall scientific message.

I therefore am confident that main concerns raised by the reviewers have been addressed and the paper can be published.

Regards,

Prof. Dr. Alexandre Langlois

Associate editor, *The Cryosphere*